# Effects of global and domestic tobacco control policies on cigarette consumption per capita: an evaluation using monthly data in China

Xiaoxin Xu,[1] Xiulan Zhang,[1] Teh Wei Hu,[2] Leonard S Miller,[3] Mengnan Xu[1]

[1]School of Social Development and Public Policy, Beijing Normal University, Beijing, China
[2]Center for International Tobacco Control, Public Health Institute, Oakland, California, USA
[3]School of Social Welfare, University of California, Berkeley, California, USA

**Correspondence to**
Dr Xiulan Zhang;
zhang99@bnu.edu.cn

## ABSTRACT

**Introduction** China consumes 44% of the world's cigarettes. Robust tobacco control measures are needed to contain the trend of increasing cigarette consumption. This paper examines the effectiveness of policy interventions introduced in China on reducing the country's tobacco use.

**Methods** The paper uses data on China's monthly cigarette consumption per capita from January 2000 to June 2017 to estimate the impact of specific policies on China's tobacco consumption. Tobacco consumption is calculated from monthly sales data from the China National Tobacco Corporation and demographic data from the China National Bureau of Statistics. The policies studied include the WHO Framework Convention on Tobacco Control (FCTC), national tobacco-related policy changes and two tobacco tax increases implemented in China during the study period. Segmented regression analysis is used to estimate the immediate effects of the policies studied and changes in the time trends resulted from these policy changes.

**Findings** The impact of national policy changes in China is almost 20 times greater than the impact of the WHO FCTC treaty itself, and national policy changes in tobacco control are a determining factor in reversing the trend of increasing tobacco consumption in China. The 2015 tax increase, which raised retail cigarette prices, produced both immediate and trend effects, with a total incremental effect 7.8 times that of the 2009 tax increase, which did not result in higher cigarette prices for the consumer.

**Interpretations** Translating global tobacco control policies into domestic policies will generate a much greater impact on reducing average cigarette consumption, and tobacco taxes that are reflected in the retail prices of cigarettes will be more effective in reducing cigarette consumption.

## Strengths and limitations of this study

► To the best of our knowledge, this study is the first systematic evaluation of the impact of both domestic and global tobacco control policies on tobacco consumption in China.

► The study compares the effectiveness of the global Framework Convention on Tobacco Control (FCTC) and domestic policies in reducing cigarette consumption in China over a period of 17.5 years.

► The data used for the policy evaluation cover the periods from no tobacco control policies in China to the implementation of FCTC policies to the changed national policies to the specific tax increases enacted in China in 2009 and 2015.

► Using the interrupted time series model, the study examines not only the immediate impact of each policy on tobacco consumption but also its impact on tobacco consumption trend.

► The limitations of this study are that the social norm change has not been incorporated into the models, and the cigarette consumption is based on wholesales rather than retails.

in the country's total cigarette consumption. In 2000, the China National Tobacco Corporation (CNTC), the state-owned tobacco monopoly, sold 76.92 billion packs of cigarettes[3]; by 2014, the number had grown to 127.48 billion packs,[4] an increase of 65.8%.

China signed the WHO Framework Convention on Tobacco Control (FCTC) in 2003; the China National People's Congress (CNPC) ratified the treaty in 2005, and China began implementing the FCTC in 2006, indicating that China's government would fulfil its legal obligation in accordance with the treaty.[5] The ratification and implementation of the FCTC provided a moral and legal high ground for advocates on tobacco control,[6] although the implementation of specific articles still had a long way to go.[7] In this paper, we refer to the WHO FCTC as an international policy. While the Chinese government has made

## INTRODUCTION

The 315 million smokers in China consume 44% of the world's cigarettes, and their average consumption is 2.3 times the world average.[1] Tobacco use increases medical expenses by billions.[1] Each year, one million people in China, many of them young, die of tobacco-related diseases.[2] China's rapid economic development in the past 40 years has been accompanied by significant growth

some efforts to control tobacco use, strong interference from the China State Tobacco Monopoly Administration (STMA), which owns CNTC and favours economic concerns over social concerns, has led to slow development and implementation of the full tobacco control policy measures.[8–13]

Between 2006 and 2015, China increased tobacco taxes twice. The first tax increase was introduced in May 2009 and was not reflected directly in cigarette retail prices. As a result, the increase had minimal immediate impact on consumers; it might have more long-term impact by changing cigarette product structure and consequent raising of average price.[14] The 2009 adjustment raised the ad valorem tax from 45% to 56% at the producer price level for class A cigarettes and from 30% to 36% for class B cigarettes. The new policy also introduced a new 5% ad valorem tax at the wholesale price level.[15 16] The intent of this 2009 adjustment was to raise government revenue from CNTC, China's tobacco producer, not to serve as a tobacco control policy instrument. Under the new scheme, the government forbade changes in the retail prices of cigarettes.[16–18] The policy was introduced primarily to counteract the impact of the global financial crisis on government revenue. Between January and April 2009, China's public revenue decreased 9.9% while public spending increased 31.7%.[19] The financial pressure prompted the government to raise the tobacco tax. In other words, this policy was driven by an economic goal, and because the policy forbade the tobacco industry from adding the tax increase to the retail price of cigarettes, the social goal was not considered at all.[16 17]

This outcome was possible because of the unique cigarette pricing mechanism under China's tobacco monopoly system. Both the cigarette allocation price, the price at which the tobacco producers offer cigarettes to the wholesalers, and the wholesale price, the price at which the wholesalers offer cigarettes to retailers, are controlled by STMA. In 2009's tobacco tax adjustment, STMA reduced the wholesale profit margin but maintained the retail price unchanged. In this sense, the 2009 tobacco tax adjustment could be regarded as a profit tax adjustment rather than an excise tax adjustment. The second tobacco tax increase occurred in May 2015. Unlike

the 2009 tax adjustment, the 2015 adjustment moved the increase from the tax base at the wholesale price level to the retail price level, a significant step away from the 2009 increase and towards China's tobacco control agenda.[20] The 2015 tax increase initiated a 0.10 RMB (US$0.0146) tax per pack at the wholesale price level and increased the ad valorem tax from 5% to 11%, a 6% point increase also at the wholesale price level. However, this time, the Chinese government allowed the tobacco industry to shift this new tax increase to the retail price, resulting in an estimated 10% increase in the retail price of cigarettes.[21]

China's new administration came into power in 2013. With support from its top leader, China's policy direction of tobacco control began to change.[8 22 23] The national policy change began with the anticorruption campaign, which was aimed at the problem of corruption within the party, state and business sectors.[24] In November 2013, the government forbade the purchase of cigarettes using public revenue. A month later, additional policies were announced that prohibited state employees/officials from smoking in public places. This significant policy initiative can be considered as a major government effort to change the social norm of smoking habits in China. Figure 1 shows the timeline of major tobacco control policies in China.

One factor that has influenced the trend of increased cigarette consumption in China is the rapid growth of personal income. China has experienced the largest economic transformation in human history. Based on data from the China National Bureau of Statistics (http://data.stats.gov.cn/ks.htm?cn=C01&zb=A0501), following the 1978 economic reform, the Chinese economy grew around 9.5% each year, becoming the second largest economy in the world according to a World Bank report.[25] In recent years, China's economic performance has remained at a relatively high level of 7% growth. While the income of people in China also has increased significantly, an increase in cigarette prices has not accompanied the gross domestic product (GDP) growth, thus making cigarettes more affordable over time.[26 27]

Waiting for China to take robust measures to control its tobacco use, change the social norm and policy landscape, reduce the institutional barriers created by STMA

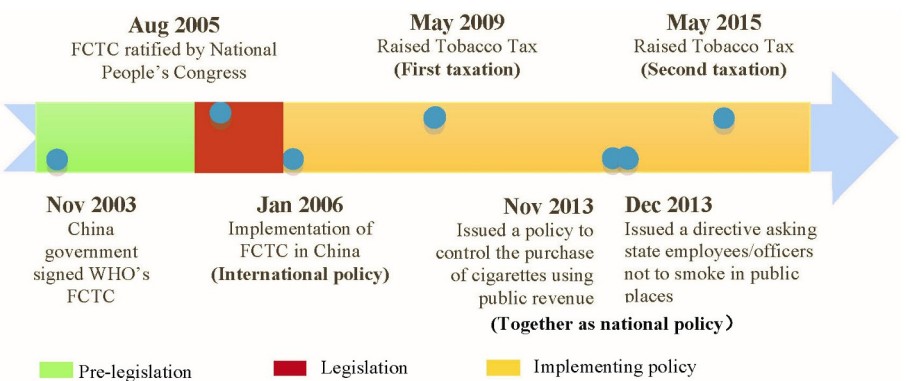

**Figure 1** Timeline of tobacco control policies in China. FCTC, Framework Convention on Tobacco Control.

and counteract the increased consumption of cigarettes resulting from income growth is a long and frustrating process.

The purpose of this paper is to estimate the relative impacts of four tobacco control policy interventions on tobacco consumption in China: first ever international public health treaty, WHO FCTC, the government's 2013 national policy forbidding using general revenue to purchase cigarettes and smoking in public by state employees/officials and the tax increases of 2009 and 2015, respectively. This is the first study to estimate the combined impact of international and domestic tobacco control policy changes on long-term trends in tobacco consumption in China.

## METHODS
### Data

We used the monthly data on cigarette sales from January 2000 to June 2017, a total of 210 months of data reported by CNTC. Sales data, collected by CNTC, are based on the purchases of retailers, so the exact monthly sales are determined by the dates when retailers buy from cigarette distributors. We extracted these data from *China Tobacco Magazine* and its website (http://www.echinatobacco.com), hosted by CNTC.

The population data were based on the China Statistical Yearbook, extracted from the website of the China National Bureau of Statistics (http://www.stats.gov.cn/tjsj/ndsj/#).

Between 2000 and 2016, China's total population increased by 9.1%. To adjust for the effect of population growth on cigarette sales, this study uses the average packs of cigarettes consumed each month per capita.[28] To estimate policy impacts, we include the GDP growth rate, the timing of policy interventions and trends initiated by each of the four policy interventions studied here.

During the study period, several tobacco control policies were implemented in China. As discussed in the Introduction section, the first was the ratification of WHO FCTC, the implementation of which began in January 2006. In May 2009, China raised cigarette taxes, but the increase was not reflected in the retail price. In 2013, tobacco control received top leadership support.

In November, a national policy was issued forbidding government funds from being used to purchase cigarettes for officials, and a month later, in December, the Central Committee of the Communist Party of China (CCCPC) and the State Council jointly issued a policy prohibiting state employees/officials from smoking in public places. This national policy targeted state officials and government agencies. In May 2015, China again raised cigarette taxes and allowed retail prices to rise.

Since the implementation of WHO FCTC, smoke-free policies have been established in different cities or regions of China. The Beijing Municipal Government passed the strictest smoke-free regulation in May 2015. But a national smoke-free law has not yet been passed. Therefore, the effect of smoke-free policies is an unmeasured effect in the model.

We divided the analysis into five time periods: (1) before the FCTC was implemented (January 2000 to December 2005); (2) between implementation of the FCTC and the first tax policy adjustment (January 2006 to April 2009); (3) between the first tax policy adjustment and implementation of the national policy change (May 2009 to October 2013); (4) between implementation of the national policy change and the second tax policy adjustment (November 2013 to April 2015) and (5) the period after all policies were implemented (May 2015 to June 2017).

Table 1 presents descriptive statistics of the GDP growth rates and average packs of cigarettes consumed during each of the five periods studied.

The GDP growth rates during the 17 years for which we have data reached 14.16% in 2007 and then dropped to 6.7% in 2016. The average growth rate over the analysis period was 9.36%. The decline in GDP growth rates began in 2012.

### Patient and public involvement

There is no patient and public involvement in data collection.

### Statistical analysis

Segmented regression analysis of interrupted time series is an effective statistical method to evaluate longitudinal effects of time-delimited interventions,[29] and it is widely

**Table 1** Average cigarette consumption and GDP growth in different periods

| Period | Policies | Number of months | Average consumption (pack/month/per capita) | Average GDP growth (%) |
|---|---|---|---|---|
| Jan 2000 to Dec 2005 | No policies | 72 | 5.66 (0.57) | 9.54 (1.06) |
| Jan 2006 to Apr 2009 | FCTC only | 40 | 6.78 (1.22) | 11.86 (2.02) |
| May 2009 to Oct 2013 | FCTC/tax1 | 54 | 7.43 (1.83) | 8.87 (1.11) |
| Nov 2013 to Apr 2015 | FCTC/tax1/national | 18 | 7.69 (2.14) | 7.32 (0.25) |
| May 2015 to Jun 2017 | FCTC/tax1/national/tax2 | 26 | 7.20 (1.56) | 6.81 (0.10) |

FCTC, Framework Convention on Tobacco Control; GDP, gross domestic product .

used in assessing policy impact especially where randomization is not feasible.[30] In this model, two parameters are estimated for each intervention studied: level and trend. The level parameter defines the y-intercept, which is the immediate effect of the intervention on the outcome. The time trend interaction with the intervention variable is the rate of change (the slope), which measures the gradual change of the outcomes due to the intervention.[30 31]

We estimated the segmented regression model using SAS AUTOREG procedures to assess the longitudinal impact of tobacco control policies on the average cigarette consumption per month per capita. We estimated levels and trends of the four interventions: FCTC (2005), first taxation (2009), national policies (2013) and second taxation (2015). The monthly pattern of sales was adjusted by the AR parameters in AUTOREG procedure.

## RESULTS

Table 2 presents an estimation of the model describing average sales of packs of cigarettes consumed per capita per month. Overall, the model is very significant with a total $R^2$ of 0.9416. The transformed $R^2$ is 0.995, indicating an extremely high fit of the model and the existence of autocorrelation.

The time effect is positive and significant, indicating the urgency to interrupt the trend of increasing cigarette consumption in China to reduce the burden of diseases and death due to smoking.

The parameter estimate shows that GDP growth has a positive and statistically significant impact on monthly cigarette consumption. GDP growth indicates a macro income effect; this effect conforms with the literature that the income elasticity of cigarette consumption is positive.

**Table 2** Autoregression model estimate of the per capita monthly cigarette consumption

| Maximum likelihood estimates of the model | | | |
|---|---|---|---|
| Log likelihood | −107.94 | Observations | 210 |
| Total $R^2$ | 0.9416 | | |
| Transformed $R^2$ | 0.9950 | | |

| Parameter estimates | | | |
|---|---|---|---|
| Variable | Estimate | t Value | Pr > \|t\| |
| Intercept | 4.8353 | 70.45 | <0.0001 |
| GDP growth | 0.0180 | 2.10 | 0.0371 |
| Time | 0.0184 | 32.99 | <0.0001 |
| FCTC | −0.1101 | −2.53 | 0.0121 |
| FCTC_time | 0.0046 | 2.37 | 0.0189 |
| Tax1 | −0.0462 | −1.38 | 0.1686 |
| Tax1_time | −0.0097 | −7.55 | <0.0001 |
| National | −0.3056 | −5.63 | <0.0001 |
| National_time | 0.0078 | 1.72 | 0.0871 |
| Tax2 | −0.4309 | −6.39 | <0.0001 |
| Tax2_time | −0.0382 | −8.30 | <0.0001 |
| AR1 | 0.6277 | 8.87 | <0.0001 |
| AR2 | 0.5907 | 8.04 | <0.0001 |
| AR3 | 0.5834 | 7.83 | <0.0001 |
| AR4 | 0.5588 | 7.33 | <0.0001 |
| AR5 | 0.5435 | 7.00 | <0.0001 |
| AR6 | 0.5458 | 6.85 | <0.0001 |
| AR7 | 0.5412 | 6.72 | <0.0001 |
| AR8 | 0.5397 | 6.72 | <0.0001 |
| AR9 | 0.5546 | 6.98 | <0.0001 |
| AR10 | 0.5662 | 7.23 | <0.0001 |
| AR11 | 0.5644 | 7.32 | <0.0001 |
| AR12 | −0.3572 | −4.83 | <0.0001 |

FCTC, Framework Convention on Tobacco Control; GDP, gross domestic product.

In terms of the impact of tobacco control policies, implementation of the FCTC in 2006 resulted in an immediate reduction in the number of cigarettes consumed. However, after the initial reduction, consumption again rose over time, indicating that either the tobacco industry developed new strategies to counteract the FCTC policy or consumers resumed the intensity of their smoking habits after the initial reaction to the macro policy change.

Similar to what happened after implementation of the international treaty, when the CCCPC and the State Council jointly issued a national policy on state employees/officials and governments in 2013, consumption of cigarettes dropped immediately. The drop following announcement of the national policy was about three times the drop in consumption after implementation of the FCTC. Again, similar to what happened after implementation of the FCTC, the trend after the change in national policy was positive subsequent to the drop, but not statistically significant. This finding shows that while the 2013 national policy changes aimed at changing cigarette-related social norms could immediately impact average sales, the after-effect was counteracted by either consumer habits or more aggressive marketing strategies by the tobacco industry.

As for China's two tobacco tax initiatives, the coefficient of the tax1 (2009) variable is not statistically significant, as one would have expected. Over time, the tobacco industry restructured its market share but the magnitude of the coefficient of the time and tax interaction term is still very small, though statistically significant. However, the coefficient of the tax2 (2015) variable and its time trend interaction term are both statistically significant with a magnitude four times larger than the FCTC effect and 40% higher than the national policy effect. As shown from the coefficient and its interaction term, the 2015 tobacco tax increase (tax2) essentially reduced per capita monthly consumption by 0.43 pack initially and then continued to reduce consumption by 0.04 pack per capita per month over time.

The implementation of both tax increases (2009 and 2015) resulted in similar initial effects and time trends. The initial effect of the second (2015) tax increase, aimed at wholesale and retail prices, was about 10 times the initial effect of the 2009 tax increase. The trend effect of the 2015 policy was about four times the trend effect of the first tax increase. In addition, unmeasured smoke-free model effects might have contributed to the big impact of the 2015 tax increase.

This finding indicates that unless specific policies are targeted at smokers, generalised policies advocating tobacco control may result in some immediate effects, but they would not be able to change the consumption of tobacco over time.

Tax increases are much more effective at changing tobacco consumption than generalised policies. This was true of even the first tax increase (in 2009), which was not reflected in the retail price of cigarettes. When the tax increase was factored into the retail price (in 2015), the impact on cigarette consumption was much larger and was sustained over time.

Figure 2 presents China's average monthly packs of cigarettes consumed per capita per year from 2000 to 2017 with and without accounting for tobacco control policies. Before the FCTC policy was implemented, between 2000 and 2005, average consumption increased from 5.1 packs to 6.3 packs per capita per month, an increase of 23.5% in 6 years.

Between 2006 and 2013, monthly cigarette consumption grew from 6.4 packs to 7.7 packs, an increase of 20.3% in 7 years. Consumption then began to decrease in 2013. By the end of 2016, the average number of packs of cigarettes consumed monthly had dropped from 7.7 to 7.2, a 6.5% decrease in 3 years. Without the tax increase, average consumption was predicted at 8.6 packs, 16.3% higher than with the tax increase.

The 2013 policy announcements by the Chinese national government changed smoking-related social norms. Combined with the global FCTC intervention and

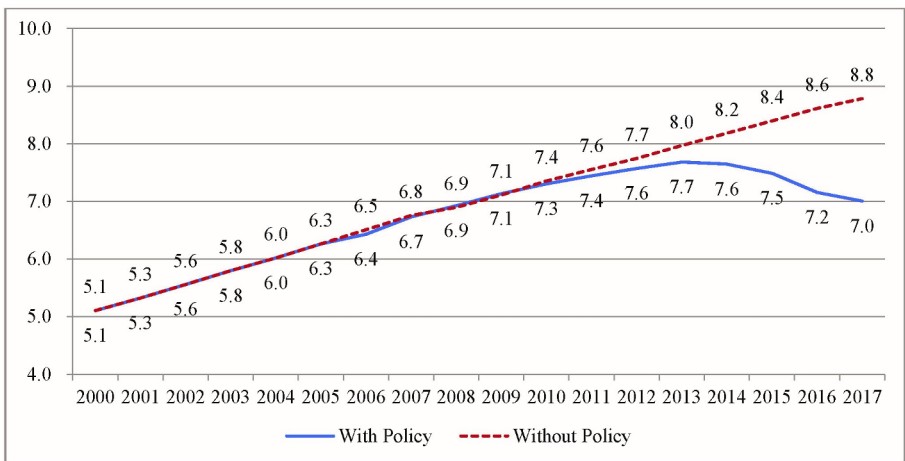

**Figure 2** Estimated monthly average packs of cigarettes consumed in China per capita with and without accounting for tobacco control policies, 2000–2017. Dotted red line (without policy), trend without tobacco control policies; blue line (with policy), trend with tobacco control policies. Both lines are predicted values from the time series model parameters.

**Table 3** Impact of tobacco control policies on average monthly cigarette consumption per capita

| Policy period | Predicted monthly consumption in packs of cigarettes with policies ($Y\hat{}_{policies=1}$) | Predicted monthly consumption in packs of cigarettes without policies ($Y\hat{}_{policies=0}$) | % Change ($Y\hat{}_{policies=1} - Y\hat{}_{policies=0}$) /$Y\hat{}_{policies=0}$ | Incremental change |
|---|---|---|---|---|
| No policies | 5.680 | 5.680 | 0.00% | 0.00% |
| FCTC only | 6.737 | 6.754 | −0.25% | −0.25% |
| FCTC/tax1 | 7.450 | 7.566 | −1.54% | −1.29% |
| FCTC/tax1/national | 7.668 | 8.202 | −6.51% | −4.97% |
| FCTC/tax1/national/tax2 | 7.176 | 8.598 | −16.54% | −10.03% |

FCTC, Framework Convention on Tobacco Control.

the first tax increase (2009), the growing trend of cigarette consumption per month per capita in China finally began to decline, and the second tax increase (2015) had a much bigger impact on the downward trend.

Based on the model estimates, we calculated the total impact of various policies on the average number of packs of cigarettes consumed per month. Table 3 presents the percentage change in average monthly consumption of cigarettes, with and without tobacco control policies. The percentage effect is calculated as follows:

$$\% \, Change = (Y\wedge_{policies=1} - Y\wedge_{policies=0}) / Y\wedge_{policies=0}$$

Table 3 shows that during the 40-month period when only the FCTC policy was in effect (January 2006 through April 2009), average cigarette consumption dropped 0.25%, due mainly to the initial impact of the FCTC. During the 54-month period that includes implementation of the FCTC and the first tax increase (May 2009 through October 2013), average consumption dropped 1.54%, and the incremental effect of the 2009 tax increase was −1.29%, indicating a very limited effect when the tax increase was not factored into the retail price.

During the 18-month period following issuance of the national policies (November 2013 through April 2015), but prior to the second tax increase instituted in May 2015, the average consumption of cigarettes dropped 6.51%, and the national policy changes alone reduced monthly consumption by 4.97%.

After the second tax increase announced in May 2015, a big decline occurred in cigarette consumption. In the 26-month period following this tax increase (May 2015 through June 2017), the average consumption of cigarettes dropped 16.54%, due mainly to the effect of the second tax increase, which alone brought down average monthly consumption by 10.03%.

Table 4 presents the predicted effects of the four policies studied on cigarette consumption measured in million packs.

Between January 2006 and June 2017, China consumed 1.348 trillion packs of cigarettes. The reduction in total consumption attributable to the policy changes was 73.6 billion packs, which is 5.18% of the sales predicted without policy interventions. Implementation of the FCTC decreased consumption by 3.5 billion packs, the first tax increase (2009) reduced consumption by 13.7 billion, the national policy announcements decreased it by 25.4 billion packs and the largest reduction came from the second tax increase (2015), including unmeasured local smoke-free policies—almost 31 billion packs in just 26 months.

From announcement of the 2013 national policy change through June 2017, China reduced the sales of cigarettes by 64.2 billion packs, a 12.57% reduction in the average consumption of cigarettes in China.

## DISCUSSION

The WHO FCTC, an international treaty, aims to provide a roadmap to address the global tobacco epidemic using effective measures and strategies. China ratified the treaty in November 2005 and began implementation in January 2006.

**Table 4** Impact of tobacco control policies on cigarette consumption in million packs

| Period | FCTC | First taxation | National | Second taxation | All policies |
|---|---|---|---|---|---|
| Jan 2006 to Apr 2009 | 893 | 0 | 0 | 0 | 893 |
| May 2009 to Oct 2013 | 1377 | 7106 | 0 | 0 | 8484 |
| Nov 2013 to Apr 2015 | 505 | 2606 | 10042 | 0 | 13153 |
| May 2015 to Jun 2017 | 771 | 3980 | 15336 | 30949 | 51036 |
| Total | 3547 | 13693 | 25377 | 30949 | 73567 |

FCTC, Framework Convention on Tobacco Control.

Calculated from table 3, this study finds that the impact of national policy changes has been almost 20 times larger than the impact of the WHO FCTC treaty itself and that national tobacco control policy changes in China have been a determining factor in reversing the increasing trend of tobacco consumption. In other words, implementing an international treaty requires national policy change to achieve the goal of reducing tobacco consumption. Ratification of the treaty alone without domestic policy implementation will have a very minimal effect.

The process of integrating global policy with domestic policy change took exactly 10 years in China (November 2003 to November 2013).[32] Our study finds that after the immediate effects of the policy changes were noted, the powerful STMA developed countermeasures to dilute the impact of the policy changes. This finding confirms the challenges faced by and the persistence required for the global and national tobacco control communities.[7 33]

Between 2006 and 2013, although the government raised the tobacco tax in May 2009, the economic goal of increasing government revenue overpowered the social goal of reducing tobacco consumption. The tax increase did not result in higher cigarette prices for the consumer, thus minimising the impact of this policy on consumption.[17 34]

When the 2015 tax policy raised retail cigarette prices, both immediate and trend effects were very significant, and the total incremental effect was 7.8 times that of the 2009 tax increase (the ratio is calculated as the incremental effect of tax2, which is the difference of FCTC/tax1/national/tax2 and FCTC/tax1/national presented in table 3: −16.54% and −6.51%=−10.03%, and the incremental effect of tax1 is the difference of FCTC/tax1 and FCTC only: −1.54% and −0.25%=−1.29%. The ratio of incremental effect tax2 and tax1 is −10.03% and −1.29%=7.8). This finding indicates that tobacco control policies should be more robust and target consumers more directly through higher prices and tougher smoke-free regulations. Because China has no national smoke-free law, and the impact of various local smoke-free regulations on national cigarette consumption is difficult to measure, the impact of taxation policy includes unmeasured effects of local smoke-free policies.

This study finds a significant positive income effect on consumption, which indicates that cigarettes have become more affordable. A recent study shows that between 2001 and 2016, the affordability of cigarettes in China increased 1.85 times. It is important to continue to raise the tobacco tax to offset the affordability influence on cigarette consumption.[27]

This study shows empirically that raising the tobacco tax through increasing retail prices is the most effective tobacco control policy instrument among the few policies implemented in China. Currently, China has a relatively low cigarette tax rate, 56% of the retail price.[20] The WHO guideline for an effective tobacco control benchmark is a tax rate of 75% of the retail price.[35] Comparing China's tax rate with the WHO guideline reveals that China has a lot of room to raise its tax on tobacco. Raising the price of cigarettes will save lives, reduce smoking-related medical costs and generate additional government revenue.

This study has some limitations. During the study period, the social norm of smoking in China has changed significantly because of the tobacco control policy changes and increased awareness of the negative health effects of smoking. This change in the social norm has not been incorporated into the model estimates. In addition, the cigarette consumption data are based on wholesale data rather than retail data. However, because the study is based on time series monthly data, and the retailers are normally not holding a large inventory, the impact of this data source on the findings is limited.

**Acknowledgements** The authors would like to thank Ms D Lynne Kaltreider for her proofreading of the manuscript.

**Contributors** XX conducted literature review, participated in data collection and manuscript writing; XZ directed and verified data collection, estimated the models and drafted the findings of the models and the discussions; TWH reviewed the models and findings and participated in writing and discussions; LSM reviewed the models and findings and participated in drafting the main findings of the models and MX participated in data collection and verification and participated in literature review.

**Funding** Bill & Melinda Gates Foundation. Role of the funding source: The funder had no role in the study design, collection, analysis or interpretation of the data, writing the manuscript or the decision to submit the paper for publication. The corresponding author had full access to all data in the study and had final responsibility for the decision to submit for publication.

**Competing interests** None declared.

**Patient consent for publication** Not required.

**Provenance and peer review** Not commissioned; externally peer reviewed.

**Data sharing statement** Extra data are available by emailing XZ.

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
