## [Reviewer comments · BMJ Open]

ARTICLE DETAILS

TITLE (PROVISIONAL)	Effects of global and domestic tobacco control policies on cigarette consumption per capita: An evaluation using monthly data in China
AUTHORS	Xu, Xiaoxin; Zhang, Xiulan; Hu, Teh-wei; Miller, Leonard; XU, Mengnan

VERSION 1 - REVIEW

REVIEWER	Anh Phuong Ngo Department of Economics, the University of Illinois at Chicago
REVIEW RETURNED	13-Aug-2018

GENERAL COMMENTS	Overall comments: The authors examined an interesting and important topic on the effects of global and domestic tobacco control policies on reducing cigarette consumption in the context of China, the country which is consuming a large share of the world's cigarette consumption. The evidence provided by the study also have important policy implications for not only China but also other countries that are in progress of adopting global tobacco control policies to protect their people from tobacco epidemic. However, there are some issues that the authors may need to work on to improve the manuscript. Please find my detailed comments on each section of the paper below. Abstract Line 10, page 3: Please state explicitly the data source that the monthly data on cigarette consumption per capita come from. Line 36, page 3: It should be "with a total marginal effect 7-8 times larger than that of the 2009 tax increase" Introduction Line 30, page 5: It would be helpful if the authors could explain in more details what the FCTC is about and what articles/guidelines China implemented after ratifying the treaty. If China just ratified the FCTC without implementing any policies recommended by FCTC, then it is obviously that the FCTC had no effects on cigarette consumption. Line 35, page 5: I think it should be "some efforts" Line 47, 48, page 5: Please explain why the 2009 tax was not reflected in cigarette retail prices directly. In other words, under what form this tax was reflected. Line 16, page 7: I think it would be clearer in terms of English to use "state employees/officers" instead of "cadres" if they are the ones that the authors referred to in the paper. Line 40, 41, page 7: Please add the citation to support the sentence ".....becoming the second largest economy in the world....."
--

	Line 43, page 7: Similarly, please add the citation to support “high level of 7% growth”. Line 11, page 8: Please specify which FCTC policy China implemented under the ratification of FCTC. Is this one policy or are they multiple policies? In addition, were they implemented at the same time or gradually one by one? Methods Line 28, page 8: Please discuss more in details about the data sources. How were the data collected? Were they reliable? Line 35, 36, page 8: How many retailers were there in the dataset? Were they mainly in urban or rural areas? In addition, were the data collected representative at the national level? Line 50, page 8: Please state explicitly how the average packs of cigarettes consumed each month per capita was derived. Line 53, page 8: Please indicate whether the data on the GDP growth rate come from. Line 20, 21, page 9: As stated in the paper, “the State Council issued a policy prohibiting cadres from smoking in public places”. Were this policy effective immediately in December or later? Line 13-24, page 9: Please discuss Table 1 in details. What do the numbers reflect about the cigarette consumption over time and the correlation between cigarette consumption and GDP growth over time? Also, it should be “pack/month/per capita”. Line 43, page 9: please add the citation to support the statement that “segmented regression analysis is widely used in assessing policy impact.” Line 50, page 15: It should be “Insert Figure 2 Here”, not “Table 2”. In addition, as shown in Figure 2, there are almost differences in the estimated monthly average packs of cigarettes consumed in China per capita with and without accounting for tobacco control policies during the 2000-2010 period. This seems to be contradictory with the findings suggested in Table 3. Line 25, page 16: Column 3, Table 3: It should be “$Y^{\text{policies}} = 0$”, not “1”. If I understand the table correctly, the authors calculated the marginal effects by subtracting the % change in previous period from the current period. In other words, the authors assume the linearity in the impacts of the policies and thus the authors may want to state this explicitly. Discussion Please discuss the limitations of the study in this section.
--	--

REVIEWER	Nigar Nargis American Cancer Society, United States
REVIEW RETURNED	12-Oct-2018

GENERAL COMMENTS	This is an important paper as it makes a very useful assessment of the impact of tobacco control policy changes on cigarette consumption in China, the largest cigarette consumer in the world. The major strength of the study is that it uses a data set that allows linking policy implementation to the sales volume by months and identification of the impact of policy changes with greater accuracy. However, the pitfall of using monthly data is that it can suffer from seasonal variation in cigarette sales, which was not accounted for in the analysis of this paper. This is one major drawback of the model which can be easily fixed in the next version.
--

	The second major comment is related to the interpretation of marginal change attributed to an individual policy. I differ with the authors with respect to the interpretation of the marginal effect of each policy in Table 3. While it is true that the effect of tobacco control policy was augmented after each new policy is introduced, the difference in the policy impact cannot be solely attributed to the new policy. It in turn dismisses the possible interaction and strengthening effect with simultaneous implementation of multiple policy measures. I suggest renaming the last column from ‘% Marginal Change’ to ‘Incremental Change’. It affects the interpretation of the impact of each policy change on cigarette consumption presented in separate columns in Table 4 as well. In my view, only the last column showing the impact of all policies is valid. I have a few other minor comments below.  1. In the Abstract, the statement “Robust tobacco control policies are needed to stop the continuing trend of increased cigarette consumption” should be part of conclusion rather than introduction. 2. Page 6 line 48: ‘6% increase’ indicates relative increase. It should be ‘6 percentage point increase’. 3. Page 7 line 26: There is no reference to Figure 1 in the text. 4. Page 8 lines 11-12: Consider rephrasing ‘the international WHO policy (FCTC)’ as ‘first ever international public health treaty WHO FCTC’. 5. In Table 1, what do the numbers in parentheses in the last two columns for Average Consumption and Average GDP Growth represent? Does the Average Consumption mean Average Per Capita Consumption? Also, is it Average GDP Growth or Average Per capita GDP Growth that you present in the last column? 6. Please explain how the per capita consumption lines presented in Figure 2 (the placeholder in the text says Insert Table 2 Here) are fitted? Are the per capita consumption levels in the line with tobacco control policies the actual observed levels? Are the per capita consumption levels in the line without tobacco control policies predicted from the estimated model of interrupted time series? It seems that, in Figure 2, the authors are attributing the difference between the line with tobacco control policies and the line without tobacco control policies (lines 28-33 in page 15) to the tax increase in 2015 only. It implies that the effects of the tobacco control policies implemented in earlier years were interrupted in 2015 when the effect of tax increase in 2015 set in. Isn’t the decline a combined effect of all the previous policy changes as you explain the following paragraph (lines 36-46 in page 15)? Please clarify the underlying assumption here. 7. The authors pointed out only the strengths and have not identified the limitations of this study.
--	--

REVIEWER	Otto Ruokolainen National Institute for Health and Welfare, Finland
REVIEW RETURNED	14-Jan-2019

GENERAL COMMENTS

Review for "Global and Domestic Tobacco Control Policies Combined Have Halted the Growing Trend of Cigarette Consumption in China: Findings from 17 Years of Monthly Data" by Xu X., Zhang X., Hu T., Miller L., Xu M. (bmjopen-2018-025092).

Thank you for the opportunity review this interesting manuscript.

Tobacco consumption in China exceeds that of the world average and the consumption has been increasing. Several policy measures are implemented in recent years aiding to curve the tobacco epidemic; however, their relative impact remains unknown. As tobacco use causes premature death and financial losses to societies, it is necessary to identify the most effective interventions to prevent the harms caused by tobacco use. The Authors examine the relative impacts of both global and domestic tobacco control policies on tobacco consumption in China with monthly data on cigarette sales from 2000 to 2017. They use segmented regression analysis in order to estimate the immediate effects as well as the trend effects resulting from these policy interventions.

The manuscript advances the knowledge on the impacts of international vs. national tobacco control policies on tobacco consumption. It is well-argued to study the interventions in China as a large proportion of the world's tobacco is being consumed there. The manuscript is well-written, uses appropriate statistical methods, present the results concisely and discuss the findings in a plausible manner. My main critiques consider the measurement/description of the FCTC implementation and lack of discussion about the possible limitations of the study.

Below are point-by-point comments on the manuscript. Some of them are merely suggestions for the Authors to decide whether to incorporate them in the manuscript or not. Some of them are also requests for clarifications to make sure I understand the presentation correctly.

Strengths and limitations of this study:

- P. 4, second bullet: Should it be "...over a period of 17.5 years"?
- There are no bullets about the possible limitations of the study (see also my comment below on the Discussion section)

Introduction:

- P. 5, lines 30–32: "...China began implementing the FCTC in 2006". It could be beneficial to explicitly state which relevant articles of the FCTC China began/has implemented (or which relevant articles China has not implemented). Some of the articles of the FCTC could be less important in the context of this study (say, article 15) than others (for example, articles 13 and 14). This would also make the interpretation of the results considering the FCTC implementation more insightful. Do the Authors see that the FCTC implementation could be also partly an "unmeasured effect in the model" (p. 9, line 39–40)?

- P. 5, line 45: "Between 2006 and 2015, China increased tobacco taxes twice". Why are these years presented as the starting and ending points? For me it seems that years 2000 and 2017 (as the data goes back from 2017 to 2000) or 2009 and 2017 (as the first

tax increase was implemented in May 2009) could be used as the starting and ending points.

Methods:

- For me, the statistical analysis section seems accurate. If needed, it could be beneficial if an expert with more segmented time series analysis experience viewed especially the Table 2, part Maximum-Likelihood Estimates for the Model.
- P. 9, line 9: "...the implementation of which began in January 2006". See comment above considering more detailed information on to which articles were (not) implemented. Also on page 9, lines 35–38, it is stated that there is no national smoke-free law in China. Does this mean that, for example, article 8 of the FCTC has not been implemented?

Results:

- P. 11, line 28: According to the Authors, some autocorrelation was present - how was this taken into account or treated?
- P. 13, lines 18–26: The Authors present credible explanations on why the consumption of tobacco rose after an initial reduction after the implementation of the FCTC. One additional reason for the increased consumption could be that the implementation of the FCTC, or parts of it, was not effective.
- P. 14, line 18: "...50 percent higher than the national policy effect". Would the Authors please elaborate on how they have come up with this proportion. If I understood correctly, the change is 41% ($\text{Tax2} / \text{National} = -0.4309 / -0.3056 = 141\%$)? Please correct me if I am mistaken.
- P. 14, lines 36–41: "In addition, unmeasured smoke-free model effects might have contributed to the big impact of the 2015 tax increase." Would the Authors please clarify why wouldn't the effects of the unmeasured smoke-free model have contributed to other interventions?
- P. 14, lines: 48: "...but they won't be able to change smokers' behavior over time." With this data it is hard to make judgement about smokers' behaviour, in my opinion. Would it be more feasible to rephrase the sentence as: "...but they won't be able to change the consumption of tobacco over time." (etc.)?
- P. 24, lines 50–52: "Tax increases are much more effective at changing smokers' behavior than generalized policies". See comment above. Also, it is unclear for me, which results the Authors refer to. Based on the Table 2, the estimates for generalized policies are FCTC = -0.1101, and National = -0.3056, whereas the estimates for tax increases are Tax1 = -0.0462 and Tax2 = -0.4309. Based on this, Tax2 > generalized policies but Tax1 < generalized policies. Please correct me if I am mistaken.
- P.16 Table 3, the column Predicted Monthly Consumption in Pack without Policies: Should it read $Y^{\text{policies}=0}$ instead of $Y^{\text{policies}=1}$?

Discussion:

- There is no discussion about the strengths and limitations of this study on the Discussion section; they are no bullet point about the limitation of this study in the manuscript after the abstract. It would be interesting and important to read what the Authors think the strengths and limitations could be (for example data based on sales vs data based on actual use/consumption).
- P. 18, line 28: "This study finds that the impact of national policy changes has been almost 20 times larger than the impact of the WHO's FCTC treaty itself" It would be, in my opinion, important to

	add this main result explicitly on the results section. At the moment, the reader may get somewhat confused on which results this is based on.  • P. 18, lines 48–53: “Our study finds that after the immediate effects of the policy changes were noted, the powerful STMA developed countermeasures to dilute the impact of the policy changes.” The current manuscript does not examine the role of the STMA, so the role of STMA could be toned down (for example, “Our study indicates...” etc.). • Page 19, second paragraph: The Authors may want to consider if it is feasible to include in this paragraph a notion that smoking is a burden on economies (or if there are opposite results from China that would also be interesting). So, the reduced consumption of tobacco would also improve economy as well as public health. • Page 19, lines 23–25: “...marginal effect was 7·8 times that of the 2009 tax increase”. As this is not explicitly stated on the results section, it could be feasible to add a reference to the Table 3 on this sentence. • P. 19, last paragraph: “This study shows empirically that raising the tobacco tax through increasing retail prices is the most effective tobacco control policy instrument in China.” This is true for the current/ examined policy interventions. However, we do not know if some other (not implemented) interventions could be as effective – or even more effective than these interventions. I would suggest that this sentence would be rephrased. • P. 20, last sentence: “Raising the tobacco tax will save lives...” Considering the effect of Tax1 and Tax2 on Table 1, it could be argued that raising the tobacco tax does not save lives but raising the price of cigarettes does (Tax2). Would it then be more precise to say that “raising the price of tobacco will save lives”? References:  • The references are appropriate and up-to-date. However, it could be useful to consider whether some more behavioural studies, for example from the ITC project, would be relevant in the Discussion section (for example https://doi.org/10.18332/tid/83849 or http://dx.doi.org/10.1136/tobaccocontrol-2013-051057)
--	--

VERSION 1 – AUTHOR RESPONSE

Reviewer 1:

Reviewer Name: Anh Phuong Ngo

Institution and Country: Department of Economics, the University of Illinois at Chicago

Please state any competing interests or state ‘None declared’: None

Please leave your comments for the authors below

Overall comments: The authors examined an interesting and important topic on the effects of global and domestic tobacco control policies on reducing cigarette consumption in the context of China, the country which is consuming a large share of the world’s cigarette consumption. The evidence provided by the study also have important policy implications for not only China but also other countries that are in progress of adopting global tobacco control policies to protect their people from tobacco epidemic.

However, there are some issues that the authors may need to work on to improve the manuscript. Please find my detailed comments on each section of the paper below.

Abstract

1. Line 10, page 3: Please state explicitly the data source that the monthly data on cigarette consumption per capita come from.

Response: We added a brief data source to the abstract, and explained the details of the data source in the method section. The monthly data on cigarette sales were collected by China National Tobacco Corporation (CNTC), which were based on the retailers' purchase from CNTC's local branches. The data on population were from China Statistical Yearbook, compiled by China National Bureau of Statistics. We calculated the monthly cigarette consumption based on these two data sets.

2. Line 36, page 3: It should be "with a total marginal effect 7-8 times larger than that of the 2009 tax increase"

Response: Corrected. Thanks for pointing this out.

Introduction

3. Line 30, page 5: It would be helpful if the authors could explain in more details what the FCTC is about and what articles/guidelines China implemented after ratifying the treaty. If China just ratified the FCTC without implementing any policies recommended by FCTC, then it is obviously that the FCTC had no effects on cigarette consumption.

Response: We added a paragraph to explain the process of China ratifying and implementing the treaty. The signing, ratifying and implementing the treaty means that China government should fulfill its legal obligation in accordance with the treaty. Since then the FCTC has provided a legal framework for advocating tobacco control, though there was still a long way for the implementation of specific articles. China government established an inter-ministry leading group to coordinate the implementation soon after the treaty ratified, and several cities have passed local smoke-free regulations.

4. Line 35, page 5: I think it should be "some efforts"

Response: Corrected. Thanks for point it out.

5. Line 47, 48, page 5: Please explain why the 2009 tax was not reflected in cigarette retail prices directly. In other words, under what form this tax was reflected.

Response: The cigarette pricing mechanism is unique in China under its tobacco monopoly system. Both the cigarette allocation price, the price that the tobacco producers offer to the wholesalers, and wholesale price, the price at which the wholesalers offer cigarettes to retailer, are controlled by China State Tobacco Monopoly Administration (STMA), and all of cigarette wholesalers are under direct leadership and control of STMA. In 2009 tobacco tax adjustment, STMA maintained the wholesale price and retail price unchanged but reduced the allocation-wholesale profit margin. In other words,

the increased tax was actually paid by the tobacco industry's profits instead of being shared between tobacco industry and smokers. In this sense, the 2009 tobacco tax adjustment could be regarded as a profit tax adjustment rather than an excise tax adjustment.

6. Line 16, page 7: I think it would be clearer in terms of English to use "state employees/officers" instead of "cadres" if they are the ones that the authors referred to in the paper.

Response: Yes. The term of "cadres" here referred to employees/officers of government, Communist Party of China (CPC), and some other public organizations directly controlled by government and CPC. We have replaced cadres with state employees/officers to make it clearer for readers.

7. Line 40, 41, page 7: Please add the citation to support the sentence ".....becoming the second largest economy in the world....."

Response: Added. Those data on China's GDP and its growth rate were extracted from China National Bureau of Statistics website (<http://data.stats.gov.cn/ks.htm?cn=C01&zb=A0501>). And World Bank's World Development Report 2012 and its World Development Indicators Database show that China has become the world's second largest economy since 2010.

8. Line 43, page 7: Similarly, please add the citation to support "high level of 7% growth".

Response: Added. Those data were extracted from China national bureau of statistics.

9. Line 11, page 8: Please specify which FCTC policy China implemented under the ratification of FCTC. Is this one policy or are they multiple policies? In addition, were they implemented at the same time or gradually one by one?

Response: WHO FCTC was enacted by China's People Congress as a whole, but it was implemented through a coordinating committee initially led by the State Development and Reform Commission, with members from the Ministries of Health, Foreign Affairs, Finance, Customs Administration, Business Administration, Quality Supervision Administration, and Tobacco Monopoly. In 2008, Tobacco Monopoly was under the management of the Ministry of Industry and Information Technology, then FCTC implementation coordination committee was led by this Ministry. In 2018, the Central Government of China gave the mandate of FCTC implementation to the Health Commission.

The FCTC includes a number of articles, providing a multiple-dimensional guideline for decision-makers in designing and implementing domestic tobacco control policies in China. The implementation of the FCTC could be regarded as a policy diffusion process. As for each article, it is a national policy agenda setting process. So the articles of the FCTC were implemented gradually and the progress of different articles varied greatly.

Methods

10. Line 28, page 8: Please discuss more in details about the data sources. How were the data collected? Were they reliable?

Response: We added some details about how the data were collected. The monthly sales data on cigarette were collected by China National Tobacco Corporation (CNTC). The data were published in China Tobacco Magazine, the official journal published bi-weekly by CNTC. Under China's tobacco monopoly system, cigarette production, allocation and wholesales are strictly controlled by CNTC, and all of the cigarette retailers must obtain licenses from CNTC. So the data collected by CNTC were reliable. However, as pointed out in our manuscript, CNTC monthly sales data were the amount of cigarette that the retailers purchased from CNTC's in that month instead of the amount that the consumers purchased from retailers.

Population and GDP growth data were extracted from China State Bureau of Statistics' online database. We also added a paragraph to discuss the limitation of the data source.

11. Line 35, 36, page 8: How many retailers were there in the dataset? Were they mainly in urban or rural areas? In addition, were the data collected representative at the national level?

Response: The data covered all of the cigarettes sold in China. Under China's tobacco monopoly system, all of the retailers, either in urban or rural areas, must have licenses authorized by CNTC and they purchase cigarettes directly from CNTC's branches. CNTC recorded all the transactions through its management information system and publish these data in China Tobacco Magazine, the official journal published bi-weekly by CNTC.

12. Line 50, page 8: Please state explicitly how the average packs of cigarettes consumed each month per capita was derived.

Response: The average packs of cigarettes consumed each month per capita equals monthly cigarettes sales divided by population in that year. Because only year population data is available, we used yearly population data to derive the average packs of cigarette consumed.

13. Line 53, page 8: Please indicate whether the data on the GDP growth rate come from.

Response: we used the data from China National Bureau of Statistics (from its website) and we indicated the source in the paper.

14. Line 20, 21, page 9: As stated in the paper, "the State Council issued a policy prohibiting cadres from smoking in public places". Were this policy effective immediately in December or later?

Response: Yes. The policy was effective immediately in December 2013.

15. Line 13-24, page 9: Please discuss Table 1 in details. What do the numbers reflect about the cigarette consumption over time and the correlation between cigarette consumption and GDP growth over time? Also, it should be "pack/month/per capita".

Response: Yes. It should be "pack/month/per capita". Thanks for point it out. Add details of Table 1 in manuscript.

16. Line 43, page 9: please add the citation to support the statement that “segmented regression analysis is widely used in assessing policy impact.”

Response: Added. Segmented regression of interrupted time series is considered to be an important quasi-experimental approach for evaluating interventions where randomization isn't feasible.

17. Line 50, page 15: It should be “Insert Figure 2 Here”, not “Table 2”. In addition, as shown in Figure 2, there are almost differences in the estimated monthly average packs of cigarettes consumed in China per capita with and without accounting for tobacco control policies during the 2000-2010 period. This seems to be contradictory with the findings suggested in Table 3.

Response: Yes. It should be “Insert Figure 2 here”, not “Table 2”. Corrected. During 2000-2005 period there were no tobacco control policies taking into account, so there were no difference in estimated monthly average cigarettes consumption per capita with or without accounting for tobacco control policies. During 2006-2009, there were very small differences with or without accounting for the tobacco control policies, since the only policy intervention in that period, FCTC, had very small effect on reducing monthly cigarettes consumption per capita. The subtle effect of FCTC could be observed if monthly cigarette consumption per capita were displayed with two digitals after decimal point.

18. Line 25, page 16: Column 3, Table 3: It should be “ $Y^{\text{policies}} = 0$ ”, not “1”. If I understand the table correctly, the authors calculated the marginal effects by subtracting the % change in previous period from the current period. In other words, the authors assume the linearity in the impacts of the policies and thus the authors may want to state this explicitly.

Response: Explained. The time-series model provides predicted outcomes with/without polies. Thanks for point it out.

Discussion

19. Please discuss the limitations of the study in this section.

Response: Added. Especially the limitation of using wholesale data taking place of retail data. We also not able to incorporate social norm change, as this study period is over a 17.5 years.

Reviewer 2

Reviewer: 2

Reviewer Name: Nigar Nargis

Institution and Country: American Cancer Society, United States

Please state any competing interests or state 'None declared': None declared.

Please leave your comments for the authors below

This is an important paper as it makes a very useful assessment of the impact of tobacco control policy changes on cigarette consumption in China, the largest cigarette consumer in the world. The major strength of the study is that it uses a data set that allows linking policy implementation to the sales volume by months and identification of the impact of policy changes with greater accuracy. However, the pitfall of using monthly data is that it can suffer from seasonal variation in cigarette sales, which was not accounted for in the analysis of this paper. This is one major drawback of the model which can be easily fixed in the next version.

Response: Thanks for pointing this out. We did a few sensitivity analyses to control the seasonal variations in cigarette sales, and the final model used lag=12 (in the SAS autoreg procedure) and yielded the best results. In addition, the biggest seasonal variations occur during the holiday seasons, so we included an indicator variable to take into account this variation.

The second major comment is related to the interpretation of marginal change attributed to an individual policy. I differ with the authors with respect to the interpretation of the marginal effect of each policy in Table 3.

While it is true that the effect of tobacco control policy was augmented after each new policy is introduced, the difference in the policy impact cannot be solely attributed to the new policy. It in turn dismisses the possible interaction and strengthening effect with simultaneous implementation of multiple policy measures. I suggest renaming the last column from '% Marginal Change' to 'Incremental Change'. It affects the interpretation of the impact of each policy change on cigarette consumption presented in separate columns in Table 4 as well. In my view, only the last column showing the impact of all policies is valid.

Response: Thanks for pointing this out. We have made corrections in the revised manuscript.

I have a few other minor comments below.

1. In the Abstract, the statement "Robust tobacco control policies are needed to stop the continuing trend of increased cigarette consumption" should be part of conclusion rather than introduction.

Response: Changes made.

2. Page 6 line 48: '6% increase' indicates relative increase. It should be '6 percentage point increase'.

Response: Changes made. Thanks for pointing this out.

3. Page 7 line 26: There is no reference to Figure 1 in the text.

Response: Changes made.

4. Page 8 lines 11-12: Consider rephrasing ‘the international WHO policy (FCTC)’ as ‘first ever international public health treaty WHO FCTC’.

Response: Changes made.

5. In Table 1, what do the numbers in parentheses in the last two columns for Average Consumption and Average GDP Growth represent? Does the Average Consumption mean Average Per Capita Consumption? Also, is it Average GDP Growth or Average Per capita GDP Growth that you present in the last column?

Response: We changed the average consumption into average per capita consumption. As for GDP growth, it means national GDP growth rate.

6. Please explain how the per capita consumption lines presented in Figure 2 (the placeholder in the text says Insert Table 2 Here) are fitted? Are the per capita consumption levels in the line with tobacco control policies the actual observed levels? Are the per capita consumption levels in the line without tobacco control policies predicted from the estimated model of interrupted time series? It seems that, in Figure 2, the authors are attributing the difference between the line with tobacco control policies and the line without tobacco control policies (lines 28-33 in page 15) to the tax increase in 2015 only. It implies that the effects of the tobacco control policies implemented in earlier years were interrupted in 2015 when the effect of tax increase in 2015 set in. Isn't the decline a combined effect of all the previous policy changes as you explain the following paragraph (lines 36-46 in page 15)? Please clarify the underlying assumption here.

Response: In the segmented regression model, the dotted red line (above) means the time trends without any interventions (here means tobacco control policies), and the blue line (bottom) means the trends with interventions. Both lines are predicted values from the time-series model parameters.

7. The authors pointed out only the strengths and have not identified the limitations of this study.

Response: Added: used wholesales rather than retails; and no social norm change variables included.

Reviewer: 3

Reviewer Name: Otto Ruokolainen

Institution and Country: National Institute for Health and Welfare, Finland

Please state any competing interests or state ‘None declared’: None declared

Please leave your comments for the authors below

Review for "Global and Domestic Tobacco Control Policies Combined Have

Halted the Growing Trend of Cigarette Consumption in China: Findings from 17

Years of Monthly Data" by Xu X., Zhang X., Hu T., Miller L., Xu M.

(bmjopen-2018-025092).

Thank you for the opportunity review this interesting manuscript.

Tobacco consumption in China exceeds that of the world average and the consumption has been increasing. Several policy measures are implemented in recent years aiding to curve the tobacco epidemic; however, their relative impact remains unknown. As tobacco use causes premature death and financial losses to societies, it is necessary to identify the most effective interventions to prevent the harms caused by tobacco use. The Authors examine the relative impacts of both global and domestic tobacco control policies on tobacco consumption in China with monthly data on cigarette sales from 2000 to 2017. They use segmented regression analysis in order to estimate the immediate effects as well as the trend effects resulting from these policy interventions.

The manuscript advances the knowledge on the impacts of international vs. national tobacco control policies on tobacco consumption. It is well-argued to study the interventions in China as a large proportion of the world's tobacco is being consumed there. The manuscript is well-written, uses appropriate statistical methods, present the results concisely and discuss the findings in a plausible manner. My main critiques consider the measurement/description of the FCTC implementation and lack of discussion about the possible limitations of the study.

Below are point-by-point comments on the manuscript. Some of them are merely suggestions for the Authors to decide whether to incorporate them in the manuscript or not. Some of them are also requests for clarifications to make sure I understand the presentation correctly.

Strengths and limitations of this study:

- P. 4, second bullet: Should it be "...over a period of 17.5 years"?
- There are no bullets about the possible limitations of the study (see also my comment below on the Discussion section)

Response: Added: used wholesales rather than retails; and no social norm change variables included.

Introduction:

- P. 5, lines 30–32: "...China began implementing the FCTC in 2006". It could be beneficial to explicitly state which relevant articles of the FCTC China began/has implemented (or which relevant articles China has not implemented). Some of the articles of the FCTC could be less important in the context of this study (say, article 15) than others (for example, articles 13 and 14). This would also make the interpretation of the results considering the FCTC implementation more insightful. Do the Authors see that the FCTC implementation could be also partly an "unmeasured effect in the model" (p. 9, line 39–40)?

Response: Yes, FCTC includes a package of measures. China's implementation of FCTC starting in 2006 mainly indicated that the government agreed to implement tobacco control measures and began to establish a coordination committee. As for each policy, it is a national policy agenda setting

process. This paper traces the changes of specific national policies within the framework, and other impact of FCTC, such as social norms, etc, are in the unmeasured effect.

- P. 5, line 45: "Between 2006 and 2015, China increased tobacco taxes twice". Why are these years presented as the starting and ending points? For me it seems that years 2000 and 2017 (as the data goes back from 2017 to 2000) or 2009 and 2017 (as the first tax increase was implemented in May 2009) could be used as the starting and ending points.

Response: In the segmented regression model, before 2006, there was no specific national policy change, so the intervention effect is not counted. Similar to other policies, the intervention is considered as 1 in the model when intervention started and 0 when there is no intervention. The time period of 2000 to 2017 is used study the trends with and without inventions.

Methods:

- For me, the statistical analysis section seems accurate. If needed, it could be beneficial if an expert with more segmented time series analysis experience viewed especially the Table 2, part Maximum-Likelihood Estimates for the Model.

Response: Two authors of the paper Professors Hu and Miller are Berkeley health economists with many years of health economic research experience, and Professor Zhang has taught econometrics for over 15 years. We have sufficient confidence in the model.

- P. 9, line 9: "...the implementation of which began in January 2006". See comment above considering more detailed information on to which articles were (not) implemented. Also on page 9, lines 35–38, it is stated that there is no national smoke-free law in China. Does this mean that, for example, article 8 of the FCTC has not been implemented?

Response: The modelling of the policy intervention is explained in above. Yes, China has not passed a national smoke-free law.

Results:

- P. 11, line 28: According to the Authors, some autocorrelation was present - how was this taken into account or treated?

Response: we used SAS Autoreg model and used lag=12 to take into account the autocorrelation (we also did a few sensitivity analyses and took the best results in the final model).

- P. 13, lines 18–26: The Authors present credible explanations on why the consumption of tobacco rose after an initial reduction after the implementation of the FCTC. One additional reason for the increased consumption could be that the implementation of the FCTC, or parts of it, was not effective.

Response: FCTC implementation is a policy-by-policy process in China. Ratifying FCTC in 2003 didn't indicated a time frame for each policy.

• P. 14, line 18: "...50 percent higher than the national policy effect". Would the Authors please elaborate on how they have come up with this proportion. If I understood correctly, the change is 41% ($\text{Tax2} / \text{National} = -0.4309 / -0.3056 = 141\%$)? Please correct me if I am mistaken.

Response: Corrected. Thanks for pointing this out.

• P. 14, lines 36–41: "In addition, unmeasured smoke-free model effects might have contributed to the big impact of the 2015 tax increase." Would the Authors please clarify why wouldn't the effects of the unmeasured smoke-free model have contributed to other interventions?

Response: Although there is no national smoke-free law in China, several cities have passed local smoke-free laws. 2015 is a milestone in China city-level smoke-free legislation- Beijing, the capital of China, passed a smoke-free law that banned smoking in all indoor public places, workplaces, and public transport. The effects of smoke-free have not been incorporated into the model, because more than 30 cities in China that passed smoke-free laws have a large span in legislative time. We discussed this issue as un-measured effect in the paper.

• P. 14, lines: 48: "...but they won't be able to change smokers' behavior over time." With this data it is hard to make judgement about smokers' behaviour, in my opinion. Would it be more feasible to rephrase the sentence as: "...but they won't be able to change the consumption of tobacco over time." (etc.)?

Response: Corrected. Thanks for pointing this out.

• P. 24, lines 50–52: "Tax increases are much more effective at changing smokers' behavior than generalized policies". See comment above. Also, it is unclear for me, which results the Authors refer to. Based on the Table 2, the estimates for generalized policies are $\text{FCTC} = -0.1101$, and $\text{National} = -0.3056$, whereas the estimates for tax increases are $\text{Tax1} = -0.0462$ and $\text{Tax2} = -0.4309$. Based on this, $\text{Tax2} >$ generalized policies but $\text{Tax1} <$ generalized policies. Please correct me if I am mistaken.

Response: Corrected. Thanks for pointing this out. The total effect of each policy is both the intercept and the time effects.

• P.16 Table 3, the column Predicted Monthly Consumption in Pack without Policies: Should it read $Y^{\text{policies}=0}$ instead of $Y^{\text{policies}=1}$?

Response: Corrected. Thanks for pointing this out.

Discussion:

• There is no discussion about the strengths and limitations of this study on the Discussion section; they are no bullet point about the limitation of this study in the manuscript after the abstract. It would be interesting and important to read what the Authors think the strengths and limitations could be (for example data based on sales vs data based on actual use/consumption).

Response: we added limitation in the manuscript.

• P. 18, line 28: “This study finds that the impact of national policy changes has been almost 20 times larger than the impact of the WHO’s FCTC treaty itself” It would be, in my opinion, important to add this main result explicitly on the results section. At the moment, the reader may get somewhat confused on which results this is based on.

Response: we added more explanation on how the effect is calculated from table 3.

• P. 18, lines 48–53: “Our study finds that after the immediate effects of the policy changes were noted, the powerful STMA developed countermeasures to dilute the impact of the policy changes.” The current manuscript does not examine the role of the STMA, so the role of STMA could be toned down (for example, “Our study indicates...” etc.).

Response: we included more discussion on China FCTC background, and the STMA is part of the discussion for implementing FCTC.

• Page 19, second paragraph: The Authors may want to consider if it is feasible to include in this paragraph a notion that smoking is a burden on economies (or if there are opposite results from China that would also be interesting). So, the reduced consumption of tobacco would also improve economy as well as public health.

Response: adding this discussion will require more evidences and will make the paper too broad. We consider a separate paper on this topic in the future.

• Page 19, lines 23–25: “...marginal effect was 7·8 times that of the 2009 tax increase”. As this is not explicitly stated on the results section, it could be feasible to add a reference to the Table 3 on this sentence.

Response: the ratio is calculated as the marginal effect of tax2 which is the difference of FCTC/Tax1/National/Tax2 and FCTC/Tax1/National presented in table 3: -16.54% and $-6.51\% = -10.03\%$, and the marginal effect of tax1 is the difference of FCTC/Tax1 and FCTC Only: -1.54% and $-0.25\% = -1.29\%$. The ratio of marginal effect tax2 and tax1 is -10.03% and $-1.29\% = 7.8$. We added it into the manuscript.

• P. 19, last paragraph: “This study shows empirically that raising the tobacco tax through increasing retail prices is the most effective tobacco control policy instrument in China.” This is true for the current/ examined policy interventions. However, we do not know if some other (not

implemented) interventions could be as effective – or even more effective than these interventions. I would suggest that this sentence would be rephrased.

Response: Thanks for pointing out this. We changed the wording in the manuscript.

• P. 20, last sentence: “Raising the tobacco tax will save lives...” Considering the effect of Tax1 and Tax2 on Table 1, it could be argued that raising the tobacco tax does not save lives but raising the price of cigarettes does (Tax2). Would it then be more precise to say that “raising

the price of tobacco will save lives”?

Response: Thanks for pointing out this. We changed the wording in the manuscript.

References:

- The references are appropriate and up-to-date. However, it could be useful to consider whether some more behavioural studies, for example from the ITC project, would be relevant in the Discussion section (for example <https://doi.org/10.18332/tid/83849> or <http://dx.doi.org/10.1136/tobaccocontrol-2013-051057>)

Response: we added the limitation of this study on missing social norm variables. This social norm includes the behavioral issues.

VERSION 2 – REVIEW

REVIEWER	Anh Phuong Ngo Department of Economics, University of Illinois at Chicago, USA
REVIEW RETURNED	26-Feb-2019

GENERAL COMMENTS	Thanks the authors for revising the manuscript and addressing my comments. The manuscript has been improved a lot. After reviewing the manuscript the second time, I have some more minor comments in terms of wording as below. Page 4: I think it should be “compares the effectiveness of the global FCTC and domestic policies in reducing cigarette consumption in China over a period of 17.5 years. Page 4: It might be better to use “.....but also its impact on tobacco consumption trends...” Page 5: It should be “the ratification and implementation of the FCTC” Page 6: Please cite in the sentence “between January and April 2009, public revenue decreased 9.9% while public spending increased 31.7%. Page 6: It should be “retailers” instead of “retailors”. Page 7: Please check the sentence: “Support from its top leader, President Xi, began to change the policy direction of tobacco control in China.” Page 8: I think it should be “WHO FCTC”, not “WHO’s FCTC” Page 9: It should be “.....but a national smoke-free law has not been passed....” Pages 10-11: Statistical analysis: It would be nice if the authors could present the segmented regression model in this section. I would like to see the regression equation that they used to estimate the impact of the policies as well as control variables included in the model. Page 19: Please discuss in more detail the limitation that “the consumption of cigarette data are based on wholesales rather than retail data” in terms of how it affects your results.
--

REVIEWER	Nigar Nargis American Cancer Society, USA
REVIEW RETURNED	14-Mar-2019

GENERAL COMMENTS	The authors have addressed my comments thoroughly. I have no further comments.
--

REVIEWER	Otto Ruokolainen National Institute for Health and Welfare, Finland
REVIEW RETURNED	14-Feb-2019

GENERAL COMMENTS	Review for 'Effects of global and domestic tobacco control policies on cigarette consumption per capita: An evaluation using monthly data in China' by Xu X., Zhang X., Hu T., Miller L., Xu M. (bmjopen-2018-025092.R1) Thank you for the opportunity to review the revised version of the manuscript. In my opinion, the Authors have addressed the notions adequately made by all the three Reviewers. The revised manuscript now includes the improvements and clarifications proposed by the Reviewers. The only remark I have is that the manuscript should be proof-read once more. For example:  • Page 6, in the revised text, China State Tobacco Monopoly (STMA) is written in full, but the abbreviation has already been introduced earlier (Introduction, second paragraph, page 5). • Page 7: "The 2015 tax increase initiated a 0.10 RMB tax per pack at the wholesale price level" It could be beneficial to the readers if the corresponding price would be presented also (for example) in US dollars / UK pounds etc. • Page 9: An analogous style of presentation to the websites (with/without "http://...") should be used • Page 9 / Page 13: The word "cadre" is being used. • Table 3: In the text, the word "incremental" should be used instead of "marginal", align with the revised wording in the Table 3. • Discussion, page 18: "In other words, implementing an international treaty requires national policy and social norm changes to achieve the goal of reducing tobacco consumption." Would it be good to omit the "social norm changes" from this sentence? As pointed out in the limitations, social norm changes could not be incorporated in the model estimations. Maybe an additional sentence / clarification about this could be included in the text?
--

VERSION 2 – AUTHOR RESPONSE

Reviewer 1:

Reviewer Name: Anh Phuong Ngo

Institution and Country: Department of Economics, the University of Illinois at Chicago

Please state any competing interests or state 'None declared': none

Please leave your comments for the authors below

Thanks the authors for revising the manuscript and addressing my comments. The manuscript has been improved a lot. After reviewing the manuscript the second time, I have some more minor comments in terms of wording as below.

- Page 4: I think it should be “compares the effectiveness of the global FCTC and domestic policies in reducing cigarette consumption in China over a period of 17.5 years.

Response: Corrected. Thanks for pointing this out.

- Page 4: It might be better to use “... but also its impact on tobacco consumption trends...”

Response: Changes made. It's much better. Thank you.

- Page 5: It should be “the ratification and implementation of the FCTC”

Response: Corrected. Thanks.

- Page 6: Please cite in the sentence “between January and April 2009, public revenue decreased 9.9% while public spending increased 31.7%.

Response: Reference added. Those data were extracted from the official website of China's Ministry of Finance.

- Page 6: It should be “retailers” instead of “retailors”.

Response: Corrected. Thanks for pointing it out.

- Page 7: Please check the sentence:

“Support from its top leader, President Xi, began to change the policy direction of tobacco control in China.”

Response: Revised and references were added.

- Page 8: I think it should be “WHO FCTC”, not “WHO's FCTC”

Response: Corrected, both here and elsewhere. Thanks.

- Page 9: It should be “.....but a national smoke-free law has not been passed....”

Response: Corrected. Thanks for pointing this out.

• Pages 10-11: Statistical analysis: It would be nice if the authors could present the segmented regression model in this section. I would like to see the regression equation that they used to estimate the impact of the policies as well as control variables included in the model.

Response: The independent variables, i.e., the international and domestic tobacco control policies, and the control variables of the model were presented in Table 2. The SAS command for the estimation is as follow:

```
Proc AUTOREG data=cigsales;
```

```
Model cig_sales=gdpgrowth time fctc fctc_time Tax1 Tax1_time Npolicy Npolicy_time tax2 Tax2_time  
/dw=4 dwprob nlag=12 method=ml backstep archtest;
```

```
Output out=predsale1 predicted=yhatsale1 pm=trendhatsale1;
```

```
run;
```

• Page 19: Please discuss in more detail the limitation that “the consumption of cigarette data are based on wholesales rather than retail data” in terms of how it affects your results.

Response: The retailers purchase cigarettes from wholesalers, who are controlled by the China Tobacco Monopoly Administration. The results might be a time lag of the total consumption between the retailers and wholesalers; however, we believe the time lag has little impact on the results, as this is time series data, and retailers are unlikely to hold a big inventory.

Reviewer 2

Reviewer Name: Nigar Nargis

Institution and Country: American Cancer Society, USA

Please state any competing interests or state ‘None declared’: None declared

Please leave your comments for the authors below

The authors have addressed my comments thoroughly. I have no further comments.

Response: Thanks for the constructive comments on an earlier version of the manuscript.

Reviewer: 3

Reviewer Name: Otto Ruokolainen

Institution and Country: National Institute for Health and Welfare, Finland

Please state any competing interests or state ‘None declared’: None declared

Please leave your comments for the authors below

Review for 'Effects of global and domestic tobacco control policies on cigarette consumption per capita: An evaluation using monthly data in China' by Xu X., Zhang X., Hu T., Miller L., Xu M. (bmjopen-2018-025092.R1)

Thank you for the opportunity to review the revised version of the manuscript. In my opinion, the Authors have addressed the notions adequately made by all the three Reviewers. The revised manuscript now includes the improvements and clarifications proposed by the Reviewers. The only remark I have is that the manuscript should be proofread once more. For example:

Response: The authors have proofread the manuscript and corrected the errors in language including those listed as below.

- Page 6, in the revised text, China State Tobacco Monopoly (STMA) is written in full, but the abbreviation has already been introduced earlier (Introduction, second paragraph, page 5).

Response: Corrected. Thanks.

- Page 7: "The 2015 tax increase initiated a 0.10 RMB tax per pack at the wholesale price level" It could be beneficial to the readers if the corresponding price would be presented also (for example) in US dollars / UK pounds etc.

Response: Added, according to the exchange rate of RMB to US dollar on January 1, 2019, i.e. 1 RMB=0.146 US dollars.

- Page 9: An analogous style of presentation to the websites (with/without "http://...") should be used

Response: Changes made. Thank you.

- Page 9 / Page 13: The word "cadre" is being used.

Response: "Cadre" here and elsewhere were replaced with "state employees/officials".

- Table 3: In the text, the word "incremental" should be used instead of "marginal", align with the revised wording in the Table 3.

Response: Changes made. Thanks.

- Discussion, page 18: "In other words, implementing an international treaty requires national policy and social norm changes to achieve the goal of reducing tobacco consumption." Would it be good to omit the "social norm changes" from this sentence? As pointed out in the limitations, social norm changes could not be incorporated in the model estimations. Maybe an additional sentence / clarification about this could be included in the text?

Response: Corrected. "social norm changes" was omitted from the sentence. Thanks.